# Left Nonrecurrent Laryngeal Nerve with Situs Inversus Totalis

**DOI:** 10.3390/diagnostics12030730

**Published:** 2022-03-17

**Authors:** Yin-Yang Chen, Chi-You Liao, Chung-Chin Yao

**Affiliations:** 1Institute of Medicine, Chung Shan Medical University, Taichung City 40201, Taiwan; jeff80329@hotmail.com (Y.-Y.C.); channelb622@gmail.com (C.-Y.L.); 2Department of Surgery, Chung Shan Medical University Hospital, Taichung City 40201, Taiwan

**Keywords:** nonrecurrent laryngeal nerve, situs inversus totalis, thyroid surgery

## Abstract

The recurrent laryngeal nerve (RLN), a branch of the vagus nerve, supplies the motor and sensation function of the larynx. Generally, RLN detours around the right subclavian artery on the right side and the aortic arch on the left side. In a rare anatomical variant, called nonrecurrent laryngeal nerve (NRLN), the nerve takes an aberrant path rather than descending into the thorax as usual. First reported in 1823, NRLN is a rare anomaly arising almost exclusively on the right side, reported in 0.3–0.8% of people, and associated with vascular anomalies of embryonic aortic arch development. The atypical vascular pattern of aberrant subclavian artery (arteria lusoria) running behind the trachea and esophagus allows the vagus nerve to pass freely, which then directly branches out as NRLN at the level of the larynx. On the other hand, cases of left NRLN, only reported in 0.004% of people, are all accompanied by significant pathologies such as situs inversus totalis with opposite vascular pattern of left aberrant subclavian artery. This rare anatomical variation is clinically important, as NLRN is a major risk factor for iatrogenic injury during thyroidectomy, parathyroidectomy, and other invasive procedures in the head and neck region.

**Figure 1 diagnostics-12-00730-f001:**
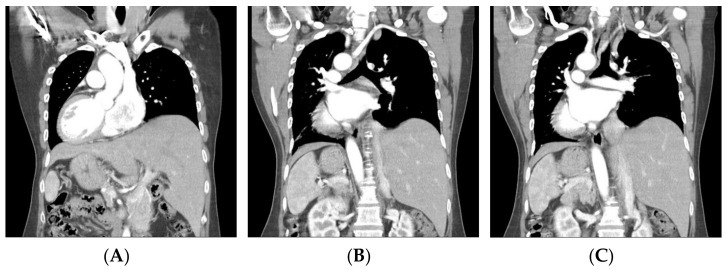
Preoperative computerized tomography (CT) revealing inversus totalis (**A**) with arterial malformation of left aberrant subclavian artery (**B**,**C**). This case concerns a 48-year-old woman who presented with a palpable left neck mass and occasional dysphagia. She had a history of thyrotoxicosis under medication control, and the last thyroid function test showed euthyroid status. She denied other systemic diseases. Physical examination showed a firm palpable nodule (about 2 cm in diameter) over the left thyroid lobe. Preoperative chest X-ray showed dextrocardia. The ultrasound scan of the neck revealed a left thyroid nodular goiter sized 2.1 × 1.7 cm^2^ with microcalcification. Fine-needle aspiration cytology was negative for malignant cells. The patient’s head and neck CT revealed not only dextrocardia with right aortic arch and aberrant left subclavian artery, but also left-sided liver and right-sided spleen, suggesting inversus totalis (Figure 1). The patient had a left lobectomy with isthmusectomy. Intraoperatively, a left-sided NRLN was dissected and preserved (Figure 2). Surgical histopathology report confirmed nodular goiter with islands of colloid-filled acini and islands of small hyperplastic acini. There was no sign of hoarseness during the postoperative period and the patient was discharged under stable condition without relative complication. In general, RLN runs into the tracheoesophageal groove, with other routes including the paratracheal area or the paraesophageal area [1]. For head and neck surgery, visualization and preservation of RLN are vital steps to prevent or minimize nerve injury. The NRLN, a rare variant of RLN, imposes a higher surgical risk of nerve injury. In the much rarer case of left NRLN, such as the case of our patient, the whole vascular and nerve development process is enantiomorphic, resulting in the inversus totalis, accompanied by left aberrant subclavian artery and left NRLN [2]. Preoperative identification of aberrant subclavian arteries and NRLNs is essential for reducing the potential risk of nerve damage. Several imaging modalities have shown their potential of preoperative detection of relevant variants. The ultrasonography pattern of the right common carotid artery at its origin level is normally seen as the “Y-sign”, since the right common carotid artery arises from the bifurcation of the brachiocephalic trunk. On the other hand, if the perioperative ultrasonography showed a right common carotid artery arising directly from the aortic arch, the right aberrant right subclavian artery and the associated right NRLN should be suspected [3]. Moreover, preoperative Computed Tomography and CT angiography are safe and convenient tools that can provide more precise images of vascular structure and promote early detection of relative anomalies of aberrant arteries and NRLN.

**Figure 2 diagnostics-12-00730-f002:**
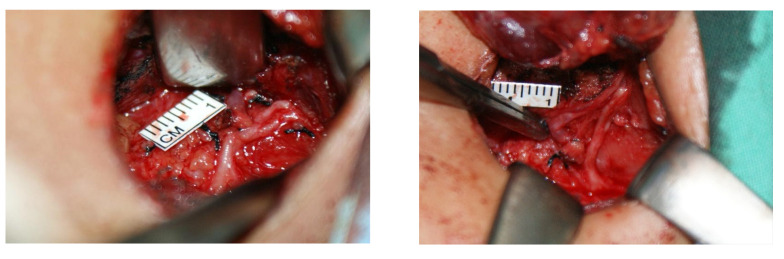
Left non-recurrent laryngeal nerve found during left thyroidectomy, arising from the vagus nerve at the level of the thyroid toward the trachea.

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
