# Peer review of "Left Nonrecurrent Laryngeal Nerve with Situs Inversus Totalis"

_diagnostics, 2022, doi:10.3390/diagnostics12030730_

Round 1

Reviewer 1 Report

It is a little difficult to recognize the positional relation of NRLN in Figure 2. Please add more explanation.

Aberrant subclavian artery may be the most important factor associated with NRLN. How often does situs inversus totalis accompany with aberrant subclavian artery? Is the frequency higher than in the cases without situs inversus totalis?

How often is NRLN seen in the cases with aberrant subclavian artery?

Reviewer 2 Report

Linguistic review of the paper is necessary. Errors are detected throughout the manuscript and the abstract. Please clarify why this publiction would be important for the reader. No refs needed in the abstract. Correct the references. 

Author Response

Journal: Diagnostics

Manuscript ID: diagnostics-1625782

Title: Left nonrecurrent laryngeal nerve with situs inversus totalis

Reviewer #2:

  1. Linguistic review of the paper is necessary. Errors are detected throughout the manuscript and the abstract. 

Response: Thanks for your suggestion and to improve this aspect. We had sent our paper for linguistic review again. We hope that these replies may meet your requirement for being published. Thank you very much for your kind assistance.

  1. Please clarify why this publication would be important for the reader?

Response: Thanks for your suggestion and to improve this aspect. This rare anatomical variation is clinically important, as NLRN is a major risk factor for iatrogenic injury during head and neck region surgery. Since most NLRN was noted in the right side and only few reports of left NRLN was presented. Hence, we describe one such case of left NRLN with situs inversus totalis, in order to raise awareness for this variation for clinical practice to avoid nerve injury. We hope that these replies may meet your requirement for being published. Thank you very much for your kind assistance.

  1. No refs needed in the abstract. Correct the references?

Response: Thanks for your suggestion and to improve this aspect. The reference in the abstract are removed and the references list is revised. We hope that these replies may meet your requirement for being published. Thank you very much for your kind assistance.
